# Sensitive Electrochemical Detection of Ammonia Nitrogen via a Platinum–Zinc Alloy Nanoflower-Modified Carbon Cloth Electrode

**DOI:** 10.3390/s24030915

**Published:** 2024-01-31

**Authors:** Guanda Wang, Guangfeng Zhou, Qingze Zhang, Dong He, Chun Zhao, Hui Suo

**Affiliations:** State Key Laboratory of Integrated Optoelectronics, College of Electronic Science and Engineering, Jilin University, Changchun 130012, China; gdwang19@mails.jlu.edu.cn (G.W.); zhougf22@mails.jlu.edu.cn (G.Z.); zhangqz1919@gmail.com (Q.Z.); hedong@jlu.edu.cn (D.H.); suohui@jlu.edu.cn (H.S.)

**Keywords:** electrochemical detection, PtZn alloy nanoflower, ammonia nitrogen, self-supported electrode

## Abstract

As a common water pollutant, ammonia nitrogen poses a serious risk to human health and the ecological environment. Therefore, it is important to develop a simple and efficient sensing scheme to achieve accurate detection of ammonia nitrogen. Here, we report a simple fabrication electrode for the electrochemical synthesis of platinum–zinc alloy nanoflowers (PtZn NFs) on the surface of carbon cloth. The obtained PtZn NFs/CC electrode was applied to the electrochemical detection of ammonia nitrogen by differential pulse voltammetry (DPV). The enhanced electrocatalytic activity of PtZn NFs and the larger electrochemical active area of the self-supported PtZn NFs/CC electrode are conducive to improving the ammonia nitrogen detection performance of the sensitive electrode. Under optimized conditions, the PtZn NFs/CC electrode exhibits excellent electrochemical performance with a wide linear range from 1 to 1000 µM, a sensitivity of 21.5 μA μM^−1^ (from 1 μM to 100 μM) and a lower detection limit of 27.81 nM, respectively. PtZn NFs/CC electrodes show excellent stability and anti-interference. In addition, the fabricated electrochemical sensor can be used to detect ammonia nitrogen in tap water and lake water samples.

## 1. Introduction

Ammonia nitrogen refers to nitrogen in the form of free ammonia (NH_3_) and ammonium ions (NH_4_^+^) in water. The sources of ammonia nitrogen are very wide, including human life activities, agricultural production and industrial emissions [1]. The presence of ammonia nitrogen has an important impact on water quality and the ecological environment [2]. High concentrations of ammonia nitrogen will cause toxicity to aquatic organisms and affect their growth and reproduction [3,4]. At the same time, it also causes algal blooms, resulting in the eutrophication of water. Ammonia nitrogen in water can be converted into carcinogenic nitrite under certain conditions, and if drunk for a long time, it is extremely harmful to human health [5]. Therefore, reducing ammonia nitrogen pollution is very important to protect the water environment [6]. In order to detect the water quality problems caused by ammonia nitrogen early, it is very necessary to monitor the concentration of ammonia nitrogen in the target water quality.

At present, the commonly used detection methods of ammonia nitrogen include titration [2], Nesser’s reagent photometry [7], phenol-hypochlorite colorimetry [8], atomic absorption spectrometry [9] and electrochemical methods [10]. Among them, the electrochemical method usually does not require complex pretreatment of water samples and has the advantages of simple operation, fast detection speed and wide measurement range. The electrochemical method was more suitable for real-time monitoring. Therefore, researchers have paid attention to the research of sensitive electrodes based on nanomaterials. The Cu-MWCNT electrode prepared by Federica et al. realized the sensitive detection of ammonia nitrogen at low concentrations by the DPV method, and the effective detection range was 3~100 μM [11]. The Ag-CNT nanomaterial electrode developed by Anamaria et al. can achieve high sensitivity detection in the range of 0.2~1 mM ammonia nitrogen concentration, and the sensitivity can reach 60 μA μM^−1^ [12]. The Cu nanocubic electrode prepared by Luo et al. can detect ammonia in the range of 5.88~5.88 mM, with a detection limit of 4.7 μM [13]. These studies had good performance in the detection range, sensitivity and detection limit, but the overall performance was still slightly insufficient.

Summarizing the research in the past decade, it was not difficult to find that Pt-based electrocatalysts are still one of the most studied ammonia nitrogen-sensitive materials, which is due to the excellent detection performance of platinum-based electrochemical ammonia nitrogen sensors [5,14]. For example, in a series of platinum-metal oxide composite sensitive electrodes developed by Zhang et al., the detection range can reach 0.5–1000 μM, and the limit of detection (LOD) was as low as 37 nM [15,16]. Because of its excellent performance in ammonia nitrogen detection, Pt-based electrochemical sensors showed great potential in practical applications. However, with the in-depth study of researchers, the shortcomings of Pt-based ammonia nitrogen sensors have also been shown.

In the process of catalytic ammonia oxidation, platinum was easily poisoned by the intermediate products of ammonia decomposition, which will reduce the detection accuracy of Pt-based sensors after a longer period of operation. In addition, as a precious metal, the relatively high cost of Pt is also one of the main reasons that hinder its large-scale commercial application [17]. Overcoming its disadvantages and enhancing its advantages have become urgent research topics. At present, the alloying of Pt with non-precious metals is an effective strategy with unique advantages [18]. In one respect, due to the reduced Pt load, the cost of preparing the sensitive electrode will be reduced. In another respect, the doped metal in the alloy changes the bond length between the metals and causes compression or tensile strain effects, which significantly improves the electrocatalytic activity and stability of the electrode [19]. For these two reasons, the platinum-based alloy PtM (M = Fe, Co, Ni, Cu, Zn, etc.) has been widely used in the field of electrocatalysis, and its activity and stability have been significantly improved [20,21,22,23,24]. For example, Zhang et al. adjusted the alloying degree of bimetallic catalyst by heat treatment, indicating the PtRu alloy could inhibit the degree of CO poisoning [25]; Tran et al. enhanced oxygen reduction performance by graphene-coated PtNi nanosponges. These studies indicated that Pt alloying can significantly improve the activity and stability of electrodes. In the field of electrochemical ammonia nitrogen detection, the platinum–iridium alloy microelectrode developed by Jiang et al. has a larger detection range, up to 1–10,000 ppm [26]. The PtCu alloy nanomaterial electrode prepared by Wang et al. also showed enhanced sensitivity and a low detection limit of 8.6 nM [27]. It can be seen that the development of Pt alloy electrodes has important research value for further improving the comprehensive performance of Pt-based electrochemical sensors.

Since the atomic radius of Zn (1.34 Å) was very close to that of Pt (1.39 Å), it made it easier for Zn to match the vacancy caused by the dissolution of surface Pt atoms, effectively avoiding the deactivation of PtZn catalyst [28]. Due to the anti-Fenton reaction of Zn, compared with Fe and Co, Zn was not poisoned by hydrogen peroxide [29]. Therefore, PtZn alloys have better durability. In addition, Li et al. demonstrated by density functional theory that Zn atoms in the Pt alloy can reduce the theoretical limit potential of ammonia oxidation intermediates and promote N-H bond breakage, which is conducive to the ammonia oxidation reaction [30]. The research of Li et al. provided a theoretical basis for the advantages of excellent stability of PtZn alloy. In summary, the application of PtZn alloy nanomaterials in the detection of ammonia nitrogen will be expected.

Carbon cloth (CC) is a commonly used collector. It has excellent electrical conductivity and corrosion resistance. Depositing the active material directly on the surface of the carbon cloth is a common strategy for constructing self-supporting electrodes. In past studies, the sensitive material needed to be fixed to the collector with the help of adhesive, which would inevitably affect the overall activity of the sensitive material [31]. The self-supporting structure of the sensitive electrode can effectively avoid the adverse effects of the adhesive and improve the specific surface area so as to provide more active sites [10]. Therefore, self-supporting electrodes have greater advantages in maintaining material activity and improving detection performance.

Previously, our research group had successfully prepared PtNi electrochemical ammonia nitrogen sensors and proved that alloying had a significant positive effect on improving the performance of Pt-based electrochemical sensors [32]. In this work, the PtZn alloy-sensitive electrode was further studied. An electrochemical ammonia nitrogen-sensitive electrode with a self-supporting electrode structure was prepared by a one-step electrodeposition in situ growth strategy of the PtZn alloy nanoflower material on a CC collector. The successful synthesis of the PtZn nanoflower not only improved the electrochemical detection ability of electrodes for ammonia nitrogen but also reduced the amount of platinum metal and improved the stability. The self-supporting electrode structure further avoids the loss of active sites. The sensitivity of the electrode to ammonia nitrogen was evaluated. Figure 1 shows the synthesis process of PtZn NFs/CC electrode and the electrochemical detection process of ammonia nitrogen. Electrochemical test results showed that PtZn NFs/CC had excellent performance for sensing ammonia nitrogen. Therefore, the development of PtZn alloy-based modified electrodes is an effective strategy to improve the detection level of ammonia nitrogen.

## 2. Experimental Section

### 2.1. Chemical Reagents

Chloroplatinic acid hexahydrate (H_2_PtCl_6_·6H_2_O, AR), zinc sulfate heptahydrate (ZnSO_4_·7H_2_O, AR), ammonium chloride (NH_4_Cl, AR), potassium hydroxide (KOH, AR), hydrochloric acid (HCl, AR), oxalic acid (H_2_C_2_O_4_;, AR), sodium chloride (NaCl, AR), potassium carbonate (K_2_CO_3_, AR) and sodium bicarbonate (NaHCO_3_, AR) were purchased from Beijing Century Company (Beijing, China).

### 2.2. Pretreatment of PtZn NFs/CC Electrode

CC was tailored as rectangle of 1 cm × 0.5 cm. CC electrode was ultrasonic washed with toluene, acetone and ethanol, respectively. To enhance the hydrophilicity of the CC, the washed CC was immersed in a mixture of concentrated nitric acid and concentrated sulfuric acid for 24 h. Finally, the CC was rinsed repeatedly with deionized water, and then the CC was soaked in the deionized water for use.

The PtZn NFs/CC electrode was electrochemically deposited using a three-electrode system. CC, 1.5 cm × 1.5 cm Pt sheet and mercury oxide reference electrode (Ag/AgCl) were used as working electrode, counting electrode and reference electrode, respectively. Electrodeposition fluid was a mixture of ZnSO_4_·7H_2_O (0.5 mM) and H_2_PtCl_6_·6H_2_O (2.5 mM). PtZn NFs/CC were electrodeposited in the range of −0.8~0.6 V (vs. Ag/AgCl) for 50 cycles at a scanning rate of 50 mV s^−1^ by cyclic voltammetry (CV) method. The final product was washed with deionized water. The electrodes synthesized under the different Pt and Zn ratios in the precursor electrolyte, with 3:0, 5:1, 2:1, 1:1, 1:2 and 0:3 being named for Pt/CC, PtZn/CC-1, PtZn/CC-2, PtZn/CC-3, PtZn/CC-4, and Zn/CC, respectively. As a controlled experiment, samples with different electrodeposition cycles of 10, 20, 30, 40, 50 and 60 cycles were denoted as PtZn/CC-10 cycles, PtZn/CC-20 cycles, PtZn/CC-30 cycles and PtZn/CC-40 cycles, PtZn/CC-50 cycles and PtZn/CC-60 cycles, respectively.

### 2.3. Characterization of Catalysts

Field emission scanning electron microscope (SEM) was used to investigate the morphologies of samples (JEOL-JEM-6700F, Tokyo, Japan). Transmission electron microscopy (TEM, JEOL-JSM-7500F, Tokyo, Japan) with energy-dispersive X-ray spectroscopy (EDS, JEOL, Tokyo, Japan) was used to confirm the microstructure of samples. X-ray diffraction (XRD, Shimadzhu-6000, Tokyo, Japan) measurements were carried out with Cu Kα radiation source (λ = 1.54056 Å) at 40 kV and 30 mA (X-ray photoelectron spectroscopy (XPS, ESCALAB-250, Waltham, MA, America).

### 2.4. Electrochemical Measurements

All electrode synthesis and electrochemical experiments were carried out on the electrochemical workstation (CHI 760D, Chenhua, Shanghai, China) with the typical three-electrode system (counter electrode: 1.5 cm × 1.5 cm Pt sheet; reference electrode: Hg/HgO). The electrochemical detection of ammonia nitrogen was tested by CV and DPV. CV tested at the range from −0.8 V to 0.2 V and scanning rate of 50 mV s^−1^. DPV was measured from −0.55 V to −0.15 V at pulse amplitude of 50 mV, pulse width of 0.2 s and pulse period of 0.5 s. The electrolyte used for electrochemical test was 1 M KOH.

## 3. Results and Discussion

### 3.1. Material Characterization

Figure 1a shows the SEM image of PtZn NFs/CC. At high deposition overpotential, Pt ions were reduced into small crystal nuclei, then continued to grow along the two-dimensional structure direction and finally became nanoleaf-like. These small nanoleaves were gradually stacked up to form nanoflower clusters. Low-power imaging (Figure 1b) showed that the flower-like PtZn nanomaterials were uniformly electrodeposited on the surface of the carbon cloth.

High-resolution transmission electron microscopy (HRTEM) was used to characterize the PtZn NF catalysts after electrochemical deposition. As shown in Figure 1c, the PtZn was mainly exposed (111) crystal plane of NFs. And the lattice fringes of PtZn displayed interplanar spacings of 0.224 nm in the particle, which match well with that of the (111) planes of the fcc PtZn alloy [25]. The literature proved that the expected spacing was 0.226 nm for the (111) lattice planes of Pt nanoparticles. It can be seen that the lattice spacing of the (111) crystal face of PtZn was slightly smaller than that of Pt (111), corresponding to a lattice compression ratio of about 1%. Researchers have reported that slight lattice strains can significantly affect the activity of Pt-based catalysts [29].

Elemental atlas analysis verified the uniform distribution of Pt and Zn, as shown in Figure 1d–f. And Pt and Zn were dispersed with a Pt/Zn atomic ratio of about 7/3, according to the HRTEM-EDS distribution in Figure 1g.

To confirm the crystal structure of the PtZn alloy, bare CC, Pt/CC and PtZn NFs/CC electrodes were characterized by XRD. As shown in Figure 2a, both Pt/CC and PtZn NFs/CC had a typical face-centered cubic (fcc) structure type (PDF#04-0802) corresponding to (111), (200) and (220) crystal faces [17]. Except for the XRD peak of bare CC, there are no other diffraction peaks in the XRD spectra of PtZn NFs/CC. The results showed that the ordered PtZn alloy phase had been successfully prepared. After Zn was introduced into the Pt matrix, the characteristic peaks of PtZn NFs/CC were shifted to higher angles. For example, the (111) plane of Pt/CC and PtZn NFs/CC were assigned to 39.9° and 40.4°, respectively. The higher 2θ angle meant that the lattice of PtZn NFs/CC shrank slightly. This result was consistent with the TEM analysis from Figure 1c. The contraction can be explained by the replacement of Pt with smaller Zn atoms, resulting in a reduction in lattice spacing [33,34].

It was widely believed that changes in the electronic structure of platinum played a crucial role in increasing activity. To determine the electronic structure of Pt, Pt/CC and PtZn NFs/CC were analyzed by XPS. Figure 2b shows the Pt 4f binding energy region for two catalysts (Pt and PtZn), which can be fitted to the 4f 7/2 and 4f 5/2 peaks corresponding to the metallic and oxidation states. The relevant binding energies in PtZn alloys and Pt are shown in Table 1. It can be found that the binding energy of PtZn NFs/CC (Pt 4f 7/2 at 71.41 eV, Pt 4f 5/2 at 74.71 eV) was transferred to the high energy value compared to Pt/NC (Pt 4f 7/2 at 71.08 eV and Pt 4f 5/2 at 74.38 eV). This indicated that PtZn alloying would cause corresponding changes in electronic structure, resulting in stronger electronic structure coupling between the active sites of PtZn NFs [29].

The metallic state of Pt-base alloys was generally considered to be the main active site. According to the XPS fitting peak area, the metal state and oxidation state area were calculated, and the percentage of metal state on Pt/C and PtZn NFs/CC was 42.36% and 51.48%, respectively. The results in Table 1 show that the PtZn alloy has more metallic states. The introduction of Zn can inhibit the formation of the oxidation state of platinum.

### 3.2. Electrochemical Characteristics of PtZn NFs/CC

The calculation of the electrochemically active area, diffusion coefficient and electron transfer number of the sensitive electrode plays an important role in studying the electron transfer kinetics and interfacial electrochemical characteristics of the electrode.

Figure 3a shows CV curves of different sensitive electrodes (PtZn NFs/CC, Pt/CC and Zn/CC electrodes) in a 0.1 M KCl solution containing 5.0 mM [Fe(CN)_6_]^3−^/^4−^; solution. Among them, the peak current value of the PtZn NFs/CC electrode was obviously larger than that of other electrodes. The electrochemically active surface area (ESCA) of the electrodes can be calculated from the Randles–Sevcik equation:(1)$$I_{pa}=(2.6878\times 10^{5})n^{3/2}ACD^{1/2}v^{1/2}$$

In Equation (1), *I_pa_*, *n*, *A*, *C*, *D* and *ν* represent the oxidation peak current (*A*), the number of electrons transferred (*n* = 1), ESCA of the electrode (cm^2^), the concentration of the probe molecules (5 × 10^−6^ mol cm^−3^), the diffusion coefficient (7.60 × 10^−6^ cm^2^ s^−1^) and the scan rate (V/s), respectively. The ECSA of PtZn NFs/CC, Pt/CC and Zn/CC electrodes were calculated to be 2.885 cm^2^, 1.762 cm^2^ and 1.135 cm^2^, respectively. The results showed that the PtZn NFs/CC electrode had a larger ECSA and more electrochemically active sites, which helped to increase the charge transport pathway and promote the participation of ammonia nitrogen in the electrocatalytic reaction [35].

The diffusion coefficient (*D*) can reflect the kinetic characteristics of the electrode, and the diffusion coefficient directly affects the rate of electrochemical reaction [36]. Through consulting the relevant literature, we learned that the Cottrell equation is the following:(2)$$i=\frac{nFA\sqrt{D}C^{0}}{\sqrt{\pi t}}$$
where *i* is the current value; *n* is the number of electrons transferred during oxidation/reduction reactions; *F* is theFaraday’s constant, 96,485 C/mol; *A* is the (flat) electrode area (cm^2^); *c*^0^ is the initial concentration of analyte ammonia in mol/cm^3^; *D* is the diffusion coefficient of ammonia, expressed in cm^2^/s; and *t* is time, with a unit of s.

As shown in Figure 3b, when *t* = 0.1 s, the current values of different electrodes are *i_PtZn_* = 0.01635 A, *i_Pt_* = 0.007965 A and *i_Zn_* = 0.003067 A, respectively. By inserting the values into the equation, the diffusion coefficients were *D_PtZn_* = 4.011 × 10^−11^ cm^2^/s, *D_Pt_* = 9.514 × 10^−12^ cm^2^/s and *D_Zn_* = 1.411 × 10^−12^ cm^2^/s, respectively. The PtZn NFs/CC electrode had a larger diffusion coefficient, so its kinetic process during ammonia oxidation was faster.

The sensitivity of PtZn NFs/CC electrode to ammonia and the key reaction conditions affecting the sensitivity were analyzed. Figure 3c shows CV curves of Pt/CC, PtZn NFs/CC and Zn/CC electrodes in 1 M KOH electrolyte. Obviously, the CV curve of PtZn NFs/CC covered a larger area. According to Faraday’s first law, the amount of charge required for a redox reaction to occur on an electrode was proportional to the amount of active material on the electrode surface. Therefore, the PtZn NFs/CC electrode had a larger electrochemical surface area. In addition, two pairs of distinct peaks in the CV curve with electrodes between −0.8 V and −0.55 V were the hydrogen adsorption/desorption processes [37]. In the negative scanning curves, the peaks between −0.05 V and −0.4 V were the adsorption peaks of OH^−^ [38].

The CV curves of the Pt/CC, PtZn NFs/CC and Zn/CC electrodes in an electrolyte containing 5 mM NH_4_Cl are shown in Figure 3d. As can be seen from Figure 3d, all electrodes showed similar NH_3_ oxidation peaks between −0.55 V and −0.1 V, but the Zn/CC electrode also had much lower ammonia performance than other platinum-based electrodes. Among them, the PtZn NFs/CC electrode showed a stronger ammonia oxidation peak, and its oxidation peak current reached 2.745 mA, which was higher than that of the Pt/CC (1.79 mA) and Zn/CC (0.345 mA) electrodes. This indicated that there were more active sites for electrochemical ammoxidation on the surface of the PtZn NFs/CC electrode.

The Pt-based electrocatalyst can electrocatalyze the electrooxidation reaction of ammonia, which can decompose ammonia into molecular nitrogen (N_2_) [39]. In past studies, the mechanism of ammonia electrooxidation was well-reviewed [14]. According to Formulas (3) and (4), free ammonia was first adsorbed with the catalyst on the electrode surface to form NH_3_ (NH_3,ads_). Then, the electrocatalyst continuously catalyzed the adsorbed NH_3_ (NH_3,ads_), conducted the dehydrogenation reaction, generated various active intermediates and finally produced the product N_2_ [40].

Integral anode reaction:NH_3,aq_→NH_3,ad_(3)
2NH_3,aq_→N_2,g_ + 6H^+^ + 6e^−^(4)

From Figure 3a,b, it has been shown that the electrochemical response of the PtZn NFs/CC electrode to ammonia was higher than that of the Pt/CC electrode obtained by the same preparation method. According to previous studies, the adjustment of crystal surface spacing is the reason for improving the electrocatalytic oxidation capacity of ammonia [41]. In the previous characterization, it was proved that the introduction of Zn can play a role in regulating the crystal structure. Therefore, the alloying of Pt and Zn atoms is an important factor in improving the electrochemical response of alloy electrodes to ammonia.

In addition to the catalyst material, the electrochemical response of ammonia was also affected by key reaction conditions, such as elemental proportions and electrochemical deposition cycles. As shown in Figure 3e, when the Pt/Zn ratio in the PtZn NFs/CC electrode reached 5:1, the peak oxidation current reached the maximum (1.79 mA). With the increase of Zn content, the response of ammonia decreased to a certain extent. However, the peak current of the Pt/Zn ratio 1:2 electrode (1.41 mA) can still reach 78.68% of that of the Pt/CC electrode (1.79 mA). It can be seen that the introduction of the Zn atom played a very positive role in improving the sensitivity of platinum-based electrodes to ammonia and reducing the content of platinum. In addition, the ammonia oxidation activity of the PtZn NFs/CC electrode increased with the increase of the electrochemical deposition cycle (Figure 3f). When the electrodeposition period reached 50 cycles, the peak current value reached the maximum.

The electron transfer kinetics and interfacial properties of the PtZn NFs/CC modified electrode were studied. As shown in Figure 4a,b, the peak oxidation current (*I_pa_*) of the electrode increased linearly with the increase of the square root of the scanning rate (*ν^1/2^*), and the linear regression equation is as follows: *I_pa_* (mA) = 0.706 *ν*^1/2^ (mV s^−1^) + 0.488 (R^2^ = 0.998). Therefore, the electrooxidation of ammonia by PtZn NFs/CC electrode was a diffusion-controlled process.

According to Lavillon’s theory [42], for irreversible processes, the relationship between the peak cathode potential (*E_pc_*) and the scanning rate (*v*) can be expressed as the following:(5)$$E_{PC}=E^{0}+\frac{RT}{\alpha nF}\ln\left(\frac{RTk}{\alpha nF}\right)-\frac{RT}{\alpha nF}lnv$$

The relationship between the peak anode potential (*E_pa_*) and the scanning rate (*v*) was given by the following formula:(6)$$E_{pa}=E^{0}-\frac{RT}{\left(1-\alpha \right)nF}\ln\left(\frac{RTk}{\left(1-\alpha \right)nF}\right)+\frac{RT}{(1-\alpha )nF}lnv$$
where *E*^0^ is the formal oxidation/reduction potential, *k* is the standard rate constant of the electrochemical reaction (usually the cathode corresponds to the reduction reaction, and the anode corresponds to the oxidation reaction), *n* is the number of electrons transferred in the reaction and *α* represents the charge transfer coefficient, generally 0.5. *R, T*, and *F* have their usual meanings and, more specifically, stand for the universal gas constant (8.31 J·mol^−1^·K^−1^), the absolute temperature (298 K) and the Faraday constant (96,485.33 C·mol^−1^), respectively.

Figure 4d shows that by fitting the relationship between the peak anode potential and the Napierian logarithm of the scanning rate, the linear regression equation was *E_pa_* = 0.0166 ln*v* − 0.0687.

By substituting the known conditions into the above formula, the value of electron transfer number n of PtZn NFs/CC electrode is close to 3. It shows that three electrons were lost during the electrocatalytic oxidation of ammonia by the PtZn NFs/CC electrode [43]. We suspect the ammonia will be broken down into nitrogen and hydrogen.

Figure 4e shows the Nyquist diagram of Pt/CC, PtZn NFs/CC and Zn/CC electrodes (the illustration was the equivalent Randles circuit), and Figure 4f shows the local magnification diagram. As can be seen in the figure, the Nyquist diagram curve is approximately semi-circular in the high-frequency region, representing the control of the charge transfer process, which means the reaction rate is limited by the charge transfer process (such as electron transport or ion transport) [33]. The small diameter of the semi-arc means that the electrode had a low charge transfer resistance (Rct). According to Figure 4e,f, the Rct values of PtZn NFs/CC, Pt/CC and Zn/CC are 3.9 Ω, 5.2 Ω and 32.2 Ω, respectively. The lower charge transfer resistance can promote the interfacial electrochemical reaction. Therefore, PtZn NFs/CC electrodes can accelerate electron transfer kinetics, enhance electrocatalysis and interface performance and are conducive to the electrochemical detection of ammonia nitrogen.

### 3.3. Determination of Ammonia

The electrochemical sensitivity of PtZn NFs/CC electrode to ammonia nitrogen was detected. Figure 5a shows the CV response curves of PtZn NFs/CC in electrolytes containing different concentrations of ammonium chloride (1 M KOH with 1–700 μM NH_4_Cl). The fitting curve in Figure 5b shows that there are two linear relationships in the range from 1 μM to 700 μM: *I*_pa_ (mA) = 0.00307x + 0.277 (R^2^ = 0.992) in the range from 1 μM to 200 μM and *I*_pa_ (mA) = 0.000419x + 0.818 (R^2^ = 0.996) in the range of 200 μM to 700 μM, respectively. It can be seen from the results that when the PtZn NFs/CC electrode was tested by the CV method, the sensitivity in the low ammonia concentration was not optimistic.

The DPV method was used to detect ammonia nitrogen in PtZn NFs/CC electrodes. The ammonia concentration measured ranges from 1 μM to 1000 μM. Figure 5c shows the DPV curves of PtZn NFs/CC electrodes at different concentrations, and Figure 5d shows the fitting curves of concentration and current value. The ammonia concentration measured ranges from 1 to 1000. In this range, the PtZn NFs/CC electrode exhibited three linear fitting curves. They were *I_pa_* (μA) = 21.5x + 0.195 (R^2^ = 0.998, 1–100 μM), *I_pa_* (μA) = 5.89x + 0.295 (R^2^ = 0.999, 100–400 μM) and *I_pa_* (μA) = 2.28x + 0.430 (R^2^ = 0.998, 400–1000 μM). The results showed that the DPV method had higher sensitivity than the CV method in detecting ammonia nitrogen at low concentrations.

Based on the sensitivity of DPV for ammonia nitrogen detection at low concentrations, the limit of detection (*LOD*) and limit of quantitation (*LOQ*) were calculated [44]. The formulas are as follows:(7)$$LOD=\frac{3\sigma }{s}$$
(8)$$LOQ=\frac{10\sigma }{s}$$
where σ is the standard deviation of the blank solution, and *s* is the slope of the calibration diagram, which is the sensitivity (21.5 μA μM^−1^). The calculated *LOD* and *LOQ* of ammonia were 27.81 nM and 92.7 nM, respectively.

Table 2 shows some of the electrochemical sensors reported in recent years. Comparing the performance parameters of the PtZn NFs/CC electrode with those in Table 2, PtZn NFs/CC electrode showed the advantages of wide detection range, low *LOD* and *LOQ* and excellent sensitivity, indicating that PtZn NFs/CC sensor had a great application prospect in the detection of ammonia nitrogen.

### 3.4. Comprehensive Performance Test

To evaluate the stability of the PtZn alloy electrode, a 600 s chronocurrent analysis test was performed at −0.35 V (relative to Hg/HgO). It can be seen from Figure 6a that PtZn NFs/CC always maintain a higher current than Pt/CC and Zn/CC, which indicates that PtZn NFs/CC has excellent electrochemical stability [28,29]. In addition, as shown in Figure 6b, the CV curve of the PtZn NFs/CC electrode did not change significantly after 100 or 200 cycles of consecutive CV scans at 1 M KOH (scanning speed 50 mV s^−1^). The results showed that the electrode had good cyclic stability and the potential to be tested continuously for a long time.

The anti-interference test results are shown in Figure 6c. All the interfering substances were measured three times to study the anti-interference test. Common interfering ions (NO^2−^, SO_4_^2−^, SO_3_^2^, HCO_3_^−^, CO_3_^2−^ and C_2_O_4_^2−^) with concentrations of 100 μM were added to the electrolyte containing ammonia. The results showed that other interfering ions had little effect on the detection of ammonia. It can be seen that PtZn NFs/CC electrodes had good anti-interference performance.

Under the same conditions, PtZn NFs/CC electrodes were synthesized seven times. As shown in Figure 6c, the DPV curves of different electrodes were collected and calculated, which were tested with 1 M KOH electrolyte. The illustration was the actual DPV curves. RSD was 3.47%. These results indicated that PtZn NFs/CC also had good reproducibility.

The samples of lake water and tap water on the campus were collected as the actual test samples. After sampling, the sample was filtered by a 0.22 µM filter membrane. The practical application value of PtZn NFs/CC electrodes was evaluated, and the calibration method was used to carry out the practical test. The results are shown in Table 3. The recovered ranged from 99.70 to 105.33%, and the relative standard deviation (RSD) was less than 4.38%. The PtZn NFs/CC electrode had good performance in water with a low ammonia concentration and had potential for practical application.

## 4. Conclusions

Inspired by the literature and theory, this research group designed and synthesized PtZn NFs for ammonia nitrogen detection in water on a CC carrier. Previously, we also successfully achieved sensitive detection of ammonia nitrogen in PtNi alloy nanomaterial electrodes. These works were part of a planned series of Pt-M electrocatalysts to investigate the effects of different elements on the ammonia nitrogen detection performance of Pt-M alloy nanomaterial-modified electrodes.

In this work, the experimental results showed that PtZn alloying played an important role in improving the performance of Pt-based sensors. On the one hand, the introduction of Zn atoms changed the crystal surface spacing of the Pt crystal and also regulated the electronic structure. This made the PtZn NFs/CC have a better electrocatalytic ability and sensitive detection ability of ammonia nitrogen than pure Pt/CC. On the other hand, Zn atoms replaced part of Pt atoms, not only reducing the use of precious metal Pt but also improving the overall stability of the Pt-based electrode. It made the PtZn NFs/CC electrode have excellent anti-interference, cycle stability and repeatability. In addition, PtZn NFs/CC electrodes had a high sensitivity, low detection limit and wide detection range in the detection of ammonia nitrogen, which had great potential for practical application. This work provided a reference for the further application of alloyed electrodes in the preparation of high-performance ammonia nitrogen electrochemical sensors.

## Data Availability

Data are contained within the article.

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
