# Peer review of "Sensitive Electrochemical Detection of Ammonia Nitrogen via a Platinum–Zinc Alloy Nanoflower-Modified Carbon Cloth Electrode"

_sensors, 2024, doi:10.3390/s24030915_

Round 1

Reviewer 1 Report

Comments and Suggestions for Authors

The paper entitled “Sensitive electrochemical detection of ammonia-nitrogen via PtZn nanoflowers modified carbon cloth electrode” by Wang et al. described the development of electrochemical sensors based on PtZn nanoflowers modified carbon cloth electrode for sensitive detection of ammonia-nitrogen in water samples.  This result is interesting, but several things need to be considered before being published as follows:

1.       Just for confirmation, free ammonia (NH3) and ammonium ions (NH4+)?

2.       Line 34-36, the sentence needs to be rewritten to obtain a clear meaning

3.       The authors are suggested to revise to make a good connection between the paper citations (ex. Zhang et al. to Li et al) in the introduction section to get a better story flow

4.       The authors are suggested to discuss the employment of nanomaterials beside noble metal alloy for ammonia nitrogen detection.

5.       The authors are suggested to draw a graphical illustration of the preparation of platinum-zinc alloy and its employment as an electrochemical sensing platform for ammonia nitrogen detection. It is also advisable to draw the sensing mechanism using the proposed electrode

6.       In the experimental section, what is the difference between chemicals and materials

7.       Line 105, the sentence needs to be rewritten.

8.       Why Hg/HgO was used as a reference electrode?

9.       Which ratio of Pt and Zn was used for the characterization using SEM, HRTEM, and XPS?

10.    The authors are suggested to investigate the electrochemical active surface area (ECSA) of each electrode using Randles Sevcik equation

11.    Why the experiments to investigate the electrochemical characteristics of PtZn NFs/CC was performed in 1M KOH? Was there any pH optimization to choosing an electrolyte solution?

12.    What is the more dominant species (ammonia or ammonium ions) when this proposed sensor was employed to investigate the electrooxidation reaction of ammonia in alkaline pH?

13.    If one of the products of ammonia oxidation in alkaline conditions is nitrogen gas liberation, is it possible to observe the production of this gas during electrochemical investigation?

14.    Why the increasing amount of Zn in PtZn alloy, the response current of ammonia reduced to a certain extent? Is there any specific role from each Pt and Zn in PtZn alloy as an electrocatalyst for ammonia oxidation?

15.    In Figure 4a, what is the possible chemical reaction that occurs at -0.6 V and -0.2 V?

16.    In Figure 4b, is the calibration plot obtained from the oxidation peak current at -0.6 V or -0.2V? which oxidation potential defines the oxidation reaction of ammonia?

17.    Line 257, R2 should be R^2.

18.    What is the equivalent Randles circuit for the Nyquist plot in Figure 4c?

19.    In Figures 5b and 5d, what is the concentration range for each of the different calibration curves?

20.    The authors are suggested to employ the chronoamperometric technique to determine the diffusion coefficient (D) for each modified electrode using the Cottrell equation.

21.    Is it possible to determine the number of transferred electrons (n) and charge transfer coefficient (α) based on Laviron’s theory for each modified electrode?

22.    How many repetitions of the measurement were performed to investigate the anti-interference test as shown in Figure 6?

23.    Is there any calibration plot for the spiked concentration of ammonia to determine its concentration in real samples (lake water and tap water)?

24.    Is there any standard method that can be used to compare the calculation result of ammonia concentration in real samples using the proposed electrochemical method?

25.    I would like to suggest reading or maybe citing the work that has been published in ACS Appl. Nano. Mater. 2024, 7, 577-593, https://doi.org/10.1021/acsanm.3c04757

Comments on the Quality of English Language

English can still be improved

Author Response

A detailed reply to the reviewers’ comments

The authors would like to express their sincere appreciation to the editor and the reviewers for their diligent evaluation of our manuscript. We are truly grateful for their valuable comments and suggestions, which have greatly enhanced the quality of our research. The manuscript has undergone thorough revisions, encompassing all the suggestions and corrections put forth by the editor and the reviewers. In response to the reviewers' insightful feedback, we have provided elucidations and clarifications where necessary. The revised manuscript has been meticulously proofread and refined, with all modifications and additions clearly highlighted for easy reference. Once again, we would like to extend our heartfelt gratitude to the editor and the reviewers for their invaluable contributions to this work. A detailed reply to the reviewers’ comments is provided as follows:

Reviewers’ comments:

The paper entitled “Sensitive electrochemical detection of ammonia-nitrogen via PtZn nanoflowers modified carbon cloth electrode” by Wang et al. described the development of electrochemical sensors based on PtZn nanoflowers modified carbon cloth electrode for sensitive detection of ammonia-nitrogen in water samples.  This result is interesting, but several things need to be considered before being published as follows:

Authors’ response:

Thank you for your careful examination and precious advice Your suggestions make our work more complete, and at the same time let us get more valuable thinking. Therefore, we have done our best to carefully revise the comments you made.

Reviewer #1:

Comment 1:

Just for confirmation, free ammonia (NH3) and ammonium ions (NH4+)?

Authors’ response:

Thank you for your careful examination and precious advice. We realized that the formula for the substance should indeed be written in the introduction. We have made changes.

“Ammonia nitrogen refers to nitrogen in the form of free ammonia (NH3) and ammonium ions (NH4+) in water.”

Comment 2:

Line 34-36, the sentence needs to be rewritten to obtain a clear meaning.

Authors’ response:

Thank you for your careful examination and precious advice. We have rewritten the sentence.

“In order to detect the water quality problems caused by ammonia nitrogen early, it is very necessary to monitor the concentration of ammonia nitrogen in the target water quality.”

Comment 3:

The authors are suggested to revise to make a good connection between the paper citations (ex. Zhang et al. to Li et al) in the introduction section to get a better story flow.

Authors’ response:

Thank you for your careful examination and precious advice. We recognize that the work of other researchers cited in our previous presentation is indeed insufficiently linked to the ideas of this work. As a result, we have carefully rewritten the introduction to give the examples a better connection.

“For example, a series of platinum-metal oxide composite sensitive electrodes developed by zhang et al., the detection range can reach 0.5 - 1000 μM, the limit of detection (LOD) was as low as 37 nM [12,13]. Because of its excellent performance in ammonia nitrogen detection, Pt-based electrochemical sensors showed great potential in practical applications. However, with the in-depth study of researchers, the shortcomings of Pt-based ammonia nitrogen sensors had also been shown.

In the process of catalytic ammonia oxidation, platinum was easily poisoned by the intermediate products of ammonia decomposition, which will reduce the detection accuracy of Pt-based sensors after a longer period of operation. In addition, as a precious metal, the relatively high cost of Pt was also one of the main reasons that hinder its large-scale commercial application [14]. To overcome its disadvantages and enhance its advantages had become an urgent research topic. At present, the alloying of Pt with non-precious metals was an effective strategy with unique advantages [15]. In one respect, due to the reduced Pt load, the cost of preparing the sensitive electrode will be reduced. In another respect, the doped metal in the alloy changes the bond length between the metals and causes compression or tensile strain effects, which significantly improves the electrocatalytic activity and stability of the electrode [16]. For these two reasons, platinum-based alloy PtM (M = Fe, Co, Ni, Cu, Zn, etc.) had been widely used in the field of electrocatalysis, and its activity and stability have been significantly improved [17–21]. For example, Zhang et al. adjusted the alloying degree bimetallic catalyst by heat treatment, indicating PtRu alloy could inhibit the degree of CO poisoning [22]; Tran et al., enhanced oxygen reduction performance by graphene-coated PtNi nanosponges. These studies indicated that Pt alloying can significantly improve the activity and stability of electrodes. In the field of electrochemical ammonia nitrogen detection, the platinum-iridium alloy microelectrode developed by Jiang et al has a larger detection range, up to 1-10000 ppm [23]; The PtCu alloy nanomaterial electrode prepared by Wang et al. also showed enhanced sensitivity and a low detection limit of 8.6 nM [24]. It can be seen that the development of Pt alloy electrodes had important research value for further improving the comprehensive performance of Pt-based electrochemical sensors.

Since the atomic radius of Zn (1.34 Å) was very close to that of Pt (1.39 Å), which made it easier for Zn to match the vacancy caused by dissolution of surface Pt atoms, effectively avoiding deactivation of PtZn catalyst [25]. Due to the anti-Fenton reaction of Zn, compared with Fe and Co, Zn was not poisoned by hydrogen peroxide [26]. Therefore, PtZn alloys had better durability. In addition, li et al. demonstrated by density functional theory that Zn atoms in Pt alloy can reduce the theoretical limit potential of ammonia oxidation intermediates and promote N-H bond breakage, which is conducive to the ammonia oxidation reaction [27]. The research of Li et al. provided a theoretical basis for the advantages of excellent stability of PtZn alloy. In summary, the application of PtZn alloy nanomaterials in the detection of ammonia nitrogen will be expected.”

Comment 4:

The authors are suggested to discuss the employment of nanomaterials beside noble metal alloy for ammonia nitrogen detection.

Authors’ response:

Thank you for your careful examination and precious advice. We really lacked examples of other non-Pt-based nanomaterial sensors. We added them in the introduction.

“At present, the commonly used detection methods of ammonia nitrogen include titration [2], Nesser's reagent photometry [7], phenol-hypochlorite colorimetry [8], atomic absorption spectrometry [9] and electrochemical methods [10]. Among them, the electrochemical method usually does not require complex pretreatment of water samples, and has the advantages of simple operation, fast detection speed and wide measurement range. Electrochemical method was more suitable for real-time monitoring. Therefore, the research of sensitive electrodes based on nanomaterials had been paid attention to by researchers. The Cu-MWCNT electrode prepared by Federica et al. realized the sensitive detection of ammonia nitrogen at low concentration by DPV method, and the effective detection range was 3 ~ 100 μM [11]. The Ag-CNT nanomaterial electrode developed by Anamaria et al. can achieve high sensitivity detection in the range of 0.2 ~ 1 mM ammonia nitrogen concentration, and the sensitivity can reach 60 μA μM1 [12]. The Cu nanocubic electrode prepared by Luo et al can detect ammonia in the range of 5.88 μM ~ 5.88 mM, with a detection limit of 4.7 μM [13]. These studies had good performance in the detection range, sensitivity and detection limit, but the overall performance was still slightly insufficient.”

Comment 5:

 The authors are suggested to draw a graphical illustration of the preparation of platinum-zinc alloy and its employment as an electrochemical sensing platform for ammonia nitrogen detection. It is also advisable to draw the sensing mechanism using the proposed electrode.

Authors’ response:

Thank you for your careful examination and precious advice. Based on your suggestion, we have drawn the schematic diagram of the synthesis of the PtZn NFs/CC electrode and its sensing mechanism.

 Scheme 1. Schematic diagram of PtZn NFs/CC synthesis route and ammonia nitrogen electrochemical detection.

Comment 6:

In the experimental section, what is the difference between chemicals and materials.

Authors’ response:

Thank you for your careful examination and precious advice. We have revised "chemicals and materials" to "chemical reagents".

Comment 7:

  Line 105, the sentence needs to be rewritten.

Authors’ response:

Thank you for your careful examination and precious advice. The sentence in line 105 has been rewritten. It is currently on line 133.

“Finally, rinsed the CC repeatedly with deionized water, and then soaked the CC in the deionized water for use.”

Comment 8:

Why Hg/HgO was used as a reference electrode?

Authors’ response:

Thank you for your careful examination and precious advice. We found that there were some clerical errors. When preparing the PtZn NFs/CC electrode, we used the Ag/AgCl reference electrode instead of the Hg/HgO reference electrode (corrected in lines 136 and 139). When PtZn NFs/CC electrode is used in ammonia nitrogen detection process, Hg/HgO is used as the reference electrode. In the process of electrode preparation, the electrolyte is slightly acidic, so we choose Ag/AgCl as the reference electrode. In the electrochemical test process, the electrolyte is a strongly alkaline 1M KOH solution, so we choose Hg/HgO as the reference electrode.

Comment 9:

Which ratio of Pt and Zn was used for the characterization using SEM, HRTEM, and XPS?

Authors’ response:

Thank you for your careful examination and precious advice. We're very sorry that we didn't write the characterization clearly. We used HRTEM-EDS to characterize the ratio of Pt and Zn. We have rewritten the sentence in line 182 of the manuscript.

“Pt and Zn were dispersed with a Pt/Zn atomic ratio of about 7/3, according to the HRTEM-EDS distribution in Figure 1(g).”

Comment 10:

 The authors are suggested to investigate the electrochemical active surface area (ECSA) of each electrode using Randles Sevcik equation.

Authors’ response:

Thank you for your careful examination and precious advice. Based on your comments, we studied the ECSA of each electrode using the Randles-Sevcik equation.

“The calculation of the electrochemically active area, diffusion coefficient and electron transfer number of the sensitive electrode plays an important role in studying the electron transfer kinetics and interfacial electrochemical characteristics of the electrode.

Figure 3 (a) showed CV curves of different sensitive electrodes (PtZn NFs/CC, Pt /CC and Zn/CC electrodes) in 0.1 M KCl solution containing 5.0 mM [Fe(CN)6]3-/4-; solution. Among them, the peak current value of PtZn NFs/CC electrode was obviously larger than that of other electrodes. The electrochemically active surface area (ESCA) of the electrodes can be calculated from the Randles-Sevcik equation:

                         (1)

In equation (1), Ipa, n, A, C, D, ν represent the oxidation peak current (A), the number of electrons transferred (n = 1), ESCA of the electrode (cm2), the concentration of the probe molecules (5 × 10-6 mol cm-3), the diffusion coefficient (7.60 × 10-6 cm2 s-1), and the scan rate (V/s), respectively.  The ECSA of PtZn NFs/CC, Pt /CC and Zn/CC electrode were calculated to be 2.885 cm2, 1.762 cm2 and 1.135 cm2, respectively. The results showed that the PtZn NFs/CC electrode had a larger ECSA and more electrochemically active sites, which helped to increase the charge transport pathway and promote the participation of ammonia-nitrogen in the electrocatalytic reaction.”

Figure 3. (a) CVs of PtZn NFs/CC, Pt /CC and Zn/CC electrodes in 0.1 M KCl solution containing 5.0 mM [Fe(CN)6]3-/4-; (b) ;CV responses of the Pt/CC, PtZn NFs/CC and Zn/CC (c) in 1 M KOH and (d) in the presence of 5 mM ammonia in 1 M KOH; CV responses of (e) the electrodes with different Pt and Zn ratios (Pt/CC, PtZn/CC-1, PtZn/CC-2, PtZn/CC-3, PtZn/CC-4 and Zn/CC) and (f) the electrodes for different electrodeposition cycles (PtZn/CC-10 cycles, PtZn/CC-20 cycles, PtZn/CC-30 cycles and PtZn/CC-40 cycles, PtZn/CC-50 cycles and PtZn/CC-60 cycles) in the presence of 5 mM ammonia in 1 M KOH.

Comment 11:

Why the experiments to investigate the electrochemical characteristics of PtZn NFs/CC was performed in 1M KOH? Was there any pH optimization to choosing an electrolyte solution?

Authors’ response:

Thank you for your careful examination and precious advice.

There are several reasons why the sensor detects ammonia nitrogen at 1M KOH:

  1. The equilibrium reaction of ammonia nitrogen in solution can be expressed as: NH3 + H2O ⇌ NH4+ + OH-, and the ionization equilibrium constant of ammonia Kb is related to the pH and temperature of the solution. A 1M KOH solution has a pH of 14. Under such strongly alkaline conditions, ammonia nitrogen exists mainly in the form of free ammonia (NH3). The electrochemical response of PtZn NFs/CC sensor to free ammonia (NH3) is obvious, but the response to ammonium ion (NH4+) is not good. Therefore, the equilibrium condition in 1M KOH solution is more conducive to improving the electrochemical response of ammonia nitrogen.

In order to better answer your question, we have supplemented the CV test experiment of PtZn NFs/CC electrode on ammonium ion (NH4+) as showed in Fig 1. In 0.1M KCl solution (pH=7), ammonia nitrogen mainly exists in the form of ammonium ion (NH4+). At this time, the electrode response current to 5mM ammonium ion is only 0.660 mA. The response current in 1M KOH solution is 2.745 mA (Fig 3 a-b in the manuscript).

Fig 1. CV curves of PtZn NFs/CC electrode for different ammonia nitrogen concentrations in 0.1M KCl solution.

  1. Provide conductive medium: 1M KOH solution is a good conductive medium that can provide enough ionic conductivity to enable the sensor to work normally.
  2. The high pH of 1M KOH solution can remove some possible interfering substances and improve the anti-interference of the sensor.

To sum up, we chose 1M KOH solution as the electrolyte for the test. This is also the concentration used in most current studies.

Comment 12:

 What is the more dominant species (ammonia or ammonium ions) when this proposed sensor was employed to investigate the electrooxidation reaction of ammonia in alkaline pH?

Authors’ response:

Thank you for your careful examination and precious advice. In response to your last comment (Comment 12), we mentioned that the PtZn NFs/CC electrode has a greater electrochemical response to free ammonia (NH3), so we chose to test in 1M KOH solution. Under these conditions, free ammonia (NH3) is the more dominant species, which is what we expected.

Comment 13:

If one of the products of ammonia oxidation in alkaline conditions is nitrogen gas liberation, is it possible to observe the production of this gas during electrochemical investigation?

Authors’ response:

Thank you for your careful examination and precious advice. During the test, we did not see gas production. Because the electrode detects ammonia nitrogen faster, the DPV test can usually be completed within 30 seconds (the effective electrocatalytic ammonia potential range is between -0.5 and -0.2V), so the amount of conversion to gas is small. That's why we don't think we've seen any gas being produced.

Comment 14:

 Why the increasing amount of Zn in PtZn alloy, the response current of ammonia reduced to a certain extent? Is there any specific role from each Pt and Zn in PtZn alloy as an electrocatalyst for ammonia oxidation?

Authors’ response:

Thank you for your careful examination and precious advice. Through the work of previous researchers, we can know that slightly changing the electronic structure of Pt is conducive to improving the catalytic performance and sensing performance of Pt-based sensors. After alloying Pt with a certain proportion of Zn, the electronic structure of Pt does change (it can be seen by XRD and XPS characterization). Therefore, the improved sensitivity of PtZn/CC-1 electrode to ammonia nitrogen is in line with expectations. When the proportion of Zn in PtZn is too high, Pt no longer plays a dominant role. In Figure 3(a-b), it can be seen that pure Zn/CC is less sensitive to ammonia nitrogen. Therefore, as the Zn content continues to increase, the response capacity of the PtZn/CC electrode will tend to be the response capacity of the Zn/CC electrode. Therefore, the response current of ammonia is reduced to a certain extent.

Comment 15:

In Figure 4a, what is the possible chemical reaction that occurs at -0.6 V and -0.2 V?

Authors’ response:

Thank you for your careful examination and precious advice.

At -0.6 V and -0.2 V, ammonia underwent a continuous dehydrogenation reaction under the electrocatalysis of Pt, and the final products were N2 and H2. This reaction process was proposed by Gerischer and Mauerer and is currently widely accepted. As shown in equations 1-7, firstly, aqueous NH3 (NH3, aq) is adsorbed on the surface of Pt to form NH3, ads. Then, NH3 and ads are continuously dehydrogenated to form various active intermediates (NH2, ads, NH, ads and Nads) to form the final product N2.

NH3,aq—— NH3,ads                               (1)

NH3,ads—— NH2,ads + H+ + e-               (2)

NH2,ads—— NHads + H+ + e-                 (3)

NHx,ads + NHy,ads—— N2Hx+y,ads           (4)

N2Hx+y,ads → N2,g+(x+y)H++(x+y)e-         (5)

NHads → Nads + H+ + e-                         (6)

Overall anode reaction:

2NH3 (aq) → N2,g + 6 H+ + 6 e-              E = 0.77 V vs SHE (7)

where x= 1 or 2, y= 1 or 2

However, in view of the fact that this process is still somewhat controversial, we only list the total reaction in the manuscript.

Comment 16:

In Figure 4b, is the calibration plot obtained from the oxidation peak current at -0.6 V or -0.2V? which oxidation potential defines the oxidation reaction of ammonia?

Authors’ response:

Thank you for your careful examination and precious advice.

The value in the calibration diagram is the peak current value of the ammonia oxidation peak at different sweep speeds.

Because the reaction process is diffusion-controlled, the peak current value will increase with the increase of the sweep speed. Therefore, the value in the calibration diagram is the peak current value of the ammonia oxidation peak at different sweep speeds. Considering that the process of ammonia oxidation is a multi-step continuous reaction process, the reaction can occur in the range of -0.6V ~ -0.2V. Therefore, we cannot determine which potential determines the oxidation of ammonia. However, by analyzing the CV curve, we can see that the potential at the peak current is the reaction potential that just enters the diffusion control process. Therefore, we guess that the potential at the peak current determines the oxidation reaction of ammonia, that is, the potential is about -0.3V ~ -0.2V.

Comment 17:

Line 257, R2 should be R^2.

Authors’ response:

Thank you for your careful examination and precious advice. We have corrected the error here.

Comment 18:

What is the equivalent Randles circuit for the Nyquist plot in Figure 4c?

Authors’ response:

Thank you for your careful examination and precious advice. We have supplemented the equivalent Randles circuit of the Nyquist plot in Figure 4e.

Figure 4. (a) The CV curves of PtZn NFs/CC electrode at different scan rates (5 to 200 mV s-1) in 1 M KOH containing 5 mM NH4Cl solution and (b) linear relationship between oxidation peak current and square root of scan rate.; (c) plots of Napierian logarithm of corresponding oxidation peak current vs Napierian logarithm of scan rate; (d) plots of corresponding oxidation peak potential vs Napierian logarithm of scan rate; (e) EIS spectrum of CuO NPs/CC and bare CC electrodes in 0.1 M KCl solution contained 5.0 mM [Fe(CN)6]3−/4− (the illustration was the equivalent Randles circuit) and (f) locally enlarged image.

Comment 19:

In Figures 5b and 5d, what is the concentration range for each of the different calibration curves?

Authors’ response:

Thank you for your careful examination and precious advice. We have supplemented the concentration range for each calibration curve in Figure 5.

Figure 5. (a) CV curves of PtZn NFs/CC in the presence of different concentrations of ammonia (1 μM - 700 μM) and (b) corresponding calibration curves; (c) DPV curves of the PtZn NFs/CC electrode to different concentration of ammonia (1 μM - 1000 μM) and (d) corresponding calibration curves. 

Comment 20:

The authors are suggested to employ the chronoamperometric technique to determine the diffusion coefficient (D) for each modified electrode using the Cottrell equation.

Authors’ response:

Thank you for your careful examination and precious advice. The effect of diffusion coefficient (D) on the performance of electrochemical sensors has not been studied before. According to your suggestion, we conducted chronoamperometric tests on different electrodes. The test curve is shown in Figure 3b.

Fig 3b. Chronoamperometric curves of different electrodes

The diffusion coefficient (D)can reflect the kinetic characteristics of the electrode, and the diffusion coefficient directly affects the rate of electrochemical reaction [36]. Through consulting the relevant literature, we learned that the Cottrell equation is:

Where i was current value; n was the number of electrons transferred during oxidation/reduction reactions; F was Faraday's constant, 96485C/mol; A was (flat) electrode area (cm2); c0 was initial concentration of analyte ammonia in mol/cm3; D was Diffusion coefficient of ammonia, expressed in cm2/s; t was time, unit s.

When t = 0.1 s, the current values of different electrodes are iPtZn = 0.01635 A, iPt = 0.007965 A, iZn = 0.003067 A, respectively. By inserting the values into the equation, the diffusion coefficients were: DPtZn = 4.011×10-11 cm2/s, DPt = 9.514×10-12 cm2/s,  DZn = 1.411×10-12 cm2/s, respectively. The PtZn NFs/CC electrode had a larger diffusion coefficient, so its kinetic process during ammonia oxidation was faster.

Comment 21:

Is it possible to determine the number of transferred electrons (n) and charge transfer coefficient (α) based on Laviron’s theory for each modified electrode?

Authors’ response:

Thank you for your careful examination and precious advice. According to your suggestion, we calculated the electron transfer number (n) of the PtZn NFs/CC electrode based on Lavillon's theory.

According to Lavillon's theory, for irreversible processes, the relationship between the peak cathode potential (Epc) and the scanning rate (v) can be expressed as:

                               (3)

The relationship between the peak anode potential (Epa) and the scanning rate (v) is given by the following formula:

                    (4)

Where E0 is the formal oxidation/reduction potential, k is the standard rate constant of the electrochemical reaction (usually the cathode corresponds to the reduction reaction, the anode corresponds to the oxidation reaction), n is the number of electrons transferred in the reaction, α represents the charge transfer coefficient, generally 0.5. R, T, and F have their usual meanings and, more specifically, stand for the universal gas constant (8.31 J·mol-1 ·K-1), the absolute temperature (298 K), and the Faraday constant (96485.33 C·mol-1), respectively.

Figure 4d showed that by fitting the relationship between the peak anode potential and the Napierian logarithm of the scanning rate, the linear regression equation was Epa = 0.0166lnv - 0.0687.

By substituting the known conditions into the above formula, the value of electron transfer number n of PtZn NFs/CC electrode is close to 3. It showed that three electrons were lost during the electrocatalytic oxidation of ammonia by the PtZn NFs/CC electrode. We suspect the ammonia will be broken down into nitrogen and hydrogen.

Comment 22:

How many repetitions of the measurement were performed to investigate the anti-interference test as shown in Figure 6?

Authors’ response:

Thank you for your careful examination and precious advice. We conducted three repeated measurements for each interfering substance to investigate the anti-interference test. We have made additional explanations in the manuscript.

“All the interfering substances were measured three times to study the anti-interference test.”

Comment 23:

Is there any calibration plot for the spiked concentration of ammonia to determine its concentration in real samples (lake water and tap water)?

Authors’ response:

Thank you for your careful examination and precious advice. Figure 5(d) shows the calibration diagram of ammonia concentration. After adding different concentrations of ammonia standard solution into the blank 1M KOH electrolyte, the linear fitting curve of the relationship between ammonia concentration and peak current value was obtained by fitting. Then the ammonia response current of the electrode to the actual water sample was tested. According to the standard curve, the measurement results are converted to the concentration of ammonia nitrogen, that is, the concentration of ammonia in the actual water sample. A specific concentration of ammonia is then added to the actual water sample (increasing the ammonia concentration by 10 μM or 30 μM). According to the measured results and standard curve, the content of ammonia nitrogen in the sample was calculated again. The recovery rate of ammonia was calculated by extrapolation method. The accuracy of ammonia detection by electrode was judged by comparing the recovery rate.

Comment 24:

Is there any standard method that can be used to compare the calculation result of ammonia concentration in real samples using the proposed electrochemical method?

Authors’ response:

Thank you for your careful examination and precious advice. The content of ammonia nitrogen in the sample was determined by standard addition method.

The standard addition method can correct the influence of interfering substances that may exist in the sample on the measurement result by adding a standard solution with known concentration. By comparing the measurement results before and after the addition of the standard solution, the content of the target substance can be accurately determined.

Comment 25:

 I would like to suggest reading or maybe citing the work that has been published in ACS Appl. Nano. Mater. 2024, 7, 577-593, https://doi.org/10.1021/acsanm.3c04757

Authors’ response:

Thank you for the literature reference. When we were about to submit the paper, we did not follow up the latest literature again, which was our mistake. This paper is very important for us to improve our current work. In addition, the related performance parameters of this paper are also included in Table 2.

Table 2. Electrochemical ammonia sensors reported in the literature.

AuNPs/CC

0.001−10,000

513.577

1.0310 nM

CV

[48]

Reviewer 2 Report

Comments and Suggestions for Authors

The work entitled "Sensitive electrochemical detection of ammonium-nitrogen via PtZn nanoflowers modified carbon cloth electrode" is devoted to the synthesis and testing of new sensors based on PtZn and continues a series of works by the team of authors on the research and application of Pt-Metal type electrocatalysts. The relevance of rapid and reproducible detection of ammonia in various objects is obvious and, in addition, the self-sustaining coating greatly increases the durability of such sensors, which is of serious practical importance. The work in its current form is devoid of major inaccuracies (which were in the early version of the previous report of the authors for PtNi sensors) and in the presented form is ready for acceptance into Sensors after correcting several minor comments.

1.        Line 51. The statement “..poor stability of Pt..” it doesn't look too correct. It is widely known that Pt-electrocatalysts are among the most stable and sustainable, which cannot be said about its cheapness.

2.        Line 127. It is necessary to specify the correct name of the country of manufacture of the equipment.

3.        Figure 6(a). Correct the caption for the X-axis.

4.        Ref 37. It is necessary to provide the correct data.

5.        List of literature. The names of the journals in the links are not always correctly abbreviated.

6.        The Introduction section generally resembles a similar section in a recent article by the authors for Pt Ni sensors. It is probably necessary to diversify the description with references to other N-containing pollutants and their control.

7.        As already mentioned, it is possible that this work is one of the planned series of studies by the authors of Pt-Metal type electrocatalysts. If this is the case, it is better to point this out at the end of the Introduction section and give a reference to your recently published article in Sensors.

Author Response

A detailed reply to the reviewers’ comments

The authors would like to express their sincere appreciation to the editor and the reviewers for their diligent evaluation of our manuscript. We are truly grateful for their valuable comments and suggestions, which have greatly enhanced the quality of our research. The manuscript has undergone thorough revisions, encompassing all the suggestions and corrections put forth by the editor and the reviewers. In response to the reviewers' insightful feedback, we have provided elucidations and clarifications where necessary. The revised manuscript has been meticulously proofread and refined, with all modifications and additions clearly highlighted for easy reference. Once again, we would like to extend our heartfelt gratitude to the editor and the reviewers for their invaluable contributions to this work. A detailed reply to the reviewers’ comments is provided as follows:

Reviewers’ comments:

The work entitled "Sensitive electrochemical detection of ammonium-nitrogen via PtZn nanoflowers modified carbon cloth electrode" is devoted to the synthesis and testing of new sensors based on PtZn and continues a series of works by the team of authors on the research and application of Pt-Metal type electrocatalysts. The relevance of rapid and reproducible detection of ammonia in various objects is obvious and, in addition, the self-sustaining coating greatly increases the durability of such sensors, which is of serious practical importance. The work in its current form is devoid of major inaccuracies (which were in the early version of the previous report of the authors for PtNi sensors) and in the presented form is ready for acceptance into Sensors after correcting several minor comments.

Authors’ response:

We are fortunate to have your review comments. Prior to this review, you and two other reviewers have made very valuable suggestions on the work of PtNi sensors, making our article more accurate and complete. This work was completed after summarizing the previous experience and mistakes. We have also carefully revised and reflected on the mistakes and suggestions you pointed out this time. Thank you for your careful examination and precious advice.

Reviewer #2:

Comment 1:

 Line 51. The statement “...poor stability of Pt..” it doesn't look too correct. It is widely known that Pt-electrocatalysts are among the most stable and sustainable, which cannot be said about its cheapness.

Authors’ response:

Thank you for your careful examination and precious advice. We realize that the views expressed here are indeed imprecise and biased. Therefore, we have reformulated the view of this place.

“However, in the process of catalytic ammonia oxidation, platinum was easily poisoned by the intermediate products of ammonia decomposition, which will reduce the detection accuracy of Pt-based sensors after a longer period of operation. In addition, as a precious metal, the relatively high cost of Pt was also one of the main reasons that hinder its large-scale commercial application.”

Comment 2:

 Line 127. It is necessary to specify the correct name of the country of manufacture of the equipment.

Authors’ response:

Thank you for your careful examination and precious advice. We have corrected name of the country of manufacture of the equipment. We corrected "Amerika" to "America".

Comment 3:

 Figure 6(a). Correct the caption for the X-axis.

Authors’ response:

Thank you for your careful examination and precious advice. We have corrected the caption for the X-axis in Figure 6 (a). We corrected "Potential (V vs Hg/HgO)" to "Time (s)".

Comment 4:

Ref 37. It is necessary to provide the correct data.

Authors’ response:

Thank you for your careful examination and precious advice. In reference 37, the sensitivity of the electrode should be 8.48 μA μM-1. We have corrected the sensitivity value.

Comment 5:

 List of literature. The names of the journals in the links are not always correctly abbreviated.

Authors’ response:

Thank you for your careful examination and precious advice. We have checked and corrected the journal name errors in the link and highlighted the corrections in red.

Comment 6:

The Introduction section generally resembles a similar section in a recent article by the authors for Pt Ni sensors. It is probably necessary to diversify the description with references to other N-containing pollutants and their control.

Authors’ response:

Thank you for your careful examination and precious advice. We have modified the introduction as much as we can, hoping to further highlight the significance of studying the ammonia nitrogen sensor of Pt alloy nanomaterials. We will try to learn the writing style and variety descriptions in relevant articles to make our work more attractive.

Comment 7:

As already mentioned, it is possible that this work is one of the planned series of studies by the authors of Pt-Metal type electrocatalysts. If this is the case, it is better to point this out at the end of the Introduction section and give a reference to your recently published article in Sensors.

Authors’ response:

Thank you very much for your guidance. In the conclusion, we listed the previous PtNi sensor articles to show what we are currently working on.

“Inspired by literature and theory, this research group designed and synthesized PtZn NFs for ammonia nitrogen detection in water on CC carrier. Previously, we also successfully achieved sensitive detection of ammonia nitrogen in PtNi alloy nano-material electrodes. These works were part of a planned series of Pt-M electrocatalysts to investigate the effects of different elements on the ammonia nitrogen detection per-formance of Pt-M alloy nanomaterial modified electrodes.

In this work, the experimental results showed that PtZn alloying played an im-portant role in improving the performance of Pt-based sensors. On the one hand, the introduction of Zn atoms changed the crystal surface spacing of Pt crystal, and also regulated the electronic structure. This made PtZn NFs/CC had better electrocatalytic ability and sensitive detection ability of ammonia nitrogen than pure Pt/CC. On the other hand, Zn atoms replaced part of Pt atoms, not only reduced the use of precious metal Pt, but also improved the overall stability of the Pt-based electrode. It made the PtZn NFs/CC electrode have excellent anti-interference, cycle stability and repeatabil-ity. In addition, PtZn NFs/CC electrode had high sensitivity, low detection limit and wide detection range in the detection of ammonia nitrogen, which had great potential for practical application. This work provided a reference for the further application of alloyed electrodes in the preparation of high-performance ammonia nitrogen electro-chemical sensors.”
